# Gaussian Process-Driven History Matching for Physical Layer Parameter Estimation in Optical Fiber Communication Networks

## Josh W. Nevin and Sam Nallaperuma and Seb J. Savory

Electrical Engineering Division, Department of Engineering, University of Cambridge
9 JJ Thomson Ave, Cambridge, CB3 0FF, UK
jn399@cam.ac.uk

## Abstract

We present a methodology for the estimation of optical network physical layer parameters from signal to noise ratio via history matching. An expensive network link simulator is emulated by a Gaussian process surrogate model, which is used to estimate a set of physical layer parameters from simulated ground truth data. The a priori knowledge assumed consists of broad parameter bounds obtained from the literature and specification sheets of typical network components, and the physics-based model of the simulator. Accurate estimation of the physical layer parameters is demonstrated with a signal to noise ratio penalty of 1 dB or greater, using only 3 simulated measurements. The proposed approach is highly flexible, allowing for the calibration of any unknown simulator input from broad a priori bounds. The role of this method in the improvement of optical network modeling is discussed.

## Introduction

Optical fiber networks form the backbone of global telecommunications. The network physical layer concerns how raw bits are transmitted using the installed network equipment, including the propagation physics of the modulated laser and the physical behavior of the components. Physics-based simulators of the physical layer are critical for the design and operation of optical networks. These simulators take as an input a set of physical layer parameters that describe the performance of the network components, as well as operational parameters such as the launch power, and then output metrics of the signal quality of transmission (QoT). However, these physical layer parameters have significant uncertainties in deployed networks, which limits the accuracy of simulators (Pointurier 2021). Moreover, physical layer parameters can change with time as the components age, meaning that parameter estimation errors may increase over the network lifetime. Therefore, physical layer parameter estimation has two crucial uses. First, it improves the modeling accuracy of physics-based network simulators by reducing uncertainty in the physical layer parameters. Second, physical parameter information can be used for diagnosis of network health, as well as for building virtual network models, such as digital twins.

Methods for the estimation of physical layer parameters proposed in the literature include least-squares fitting of a physics-based model of the SNR with free parameters to measured data from a lab (Ives et al. 2017) and data from installed network monitors (Ives, Caballero, and Savory 2018). Moreover, others have utilized monitoring data to learn physical layer parameters using a number of machine learning techniques, such as Markov chain Monte Carlo (Meng et al. 2017), maximum likelihood estimation (Bouda et al. 2018), and gradient descent (Seve et al. 2018). However, several outstanding issues remain, which we address with the proposed method. For instance, some existing techniques require measurements that are taken far from the optimal operating launch power. As the QoT in optical networks has a nonlinear dependence on the signal launch power (Agrawal 2013), making such measurements means existing network services suffer a signal to noise ratio (SNR) penalty. Furthermore, the flexibility of some proposed techniques to estimate different parameters is limited, requiring significant modifications in order to estimate new parameters. Additionally, many proposed techniques rely on gradient-based approaches, which can be prone to finding local optima. Although this risk can be mitigated to some degree, for example by starting the parameter search from a range of initial conditions, a non-gradient based technique such as history matching (HM) is less susceptible to this problem. In this work we present a novel method for estimating the set of inputs to a network simulator, consisting of physical layer parameters, that agree with SNR simulations generated for a virtual optical network with a set of ground truth parameters. This technique is demonstrated with four parameters, namely the fiber attenuation coefficient $\alpha$, the fiber nonlinearity coefficient $\gamma$, the amplifier noise figure (NF) and the transceiver back-to-back SNR $\mathrm{SNR}_0$, but is general and can be applied to any simulator input.

## Method

Here we outline the proposed method for physical layer parameter estimation, covering the machine learning techniques used, the optical network link simulator and the novel estimation algorithm.

## Gaussian Process-Driven History Matching

HM is a method for the calibration of simulators, in which sets of inputs that are consistent with a set of simulated or measured ground truth outputs are identified based on a plausibility criterion (Svalova et al. 2021). For expensive simulators, HM is often performed using computationally cheap surrogate models of the simulator, such as Gaussian process emulators (GPEs), to explore the parameter space efficiently (Rana, Ertekin, and King 2018; Gardner, Lord, and Barthorpe 2020; Svalova et al. 2021).

Gaussian Processes (GPs) are machine learning models that find a predictive mean function $\bar{f}_*$ describing the mapping between a set of inputs $X$ and targets $y$, in which a kernel function is used to model the relationship between neighboring data points (Rasmussen and Williams 2006). In this work we use the squared exponential kernel function, defined by Daub (2021) as,

$$k_{SE}(x) = \exp\left(-\frac{||x_i - x_j||^2}{2l^2}\right) + \delta I \qquad (1)$$

where $||\cdot||$ represents the L2 norm of two input vectors $x_{i,j}$, $l$ is a hyper-parameter controlling the length scale of the GP, $\delta$ controls how noise is added to the covariance matrix (Daub 2021), and $I$ is an $n \times n$ identity matrix, where $n$ is the number of examples in $X$. We choose this kernel as we do not expect a priori that the target function will contain any properties requiring a more specialized kernel, such as periodicity or multiple length scales. The plausibility criterion for GP-driven HM is defined as follows. For a single set of query inputs $x_q$ and data target $y$:

$$\text{IF } y - \bar{f}_*(x_q) \leq n_\sigma \sqrt{V[f_*(x_q)]}, \ x_q \text{ is plausible}, \qquad (2)$$

where $n_\sigma$ is the maximum number of GP predictive standard deviations a query GP prediction is permitted to deviate from the ground truth data target whilst remaining plausible. In this work, we choose $n_\sigma = 3$ as the threshold for HM. Thus, as we would expect 99.7% of the simulation values to lie within 3 predictive standard deviations $\sqrt{V[f_*(x_q)]}$ of $\bar{f}_*(x_q)$ for any set of inputs $x_q$, there is a 0.3% chance of $x_q$ being falsely ruled out.

## Optical Network Link Simulator

In this work we simulate an optical network link between two nodes, and use this simulator to infer the physical behavior of the components along this link. A detailed description of the link setup is provided in the appendix. The dependence of SNR on the launch power $P$ is given by (Savory, Vincent, and Ives 2019)

$$\text{SNR} = \left(\frac{a + bP^3}{P} + \frac{1}{\text{SNR}_0}\right)^{-1}, \qquad (3)$$

where $a$ is the total linear noise power accumulated over the link which is proportional to NF, $b$ is a scalar representing the strength of the nonlinear contribution to the noise, and $\text{SNR}_0$ is the back-to-back SNR of the transceiver, meaning

Table 1: Physical layer parameters

| PARAM. | G.TRUTH | RANGE | UNIT |
|---|---|---|---|
| $\alpha$ | 0.2 | $U[0.19, 0.22]$ | dB·km$^{-1}$ |
| NF | 4.5 | $U[4.3, 4.8]$ | dB |
| $\gamma$ | 1.2 | $U[1.0, 1.5]$ | W$^{-1}$km$^{-1}$ |
| $\text{SNR}_0$ | 14.8 | $U[14.5, 15.2]$ | dB |

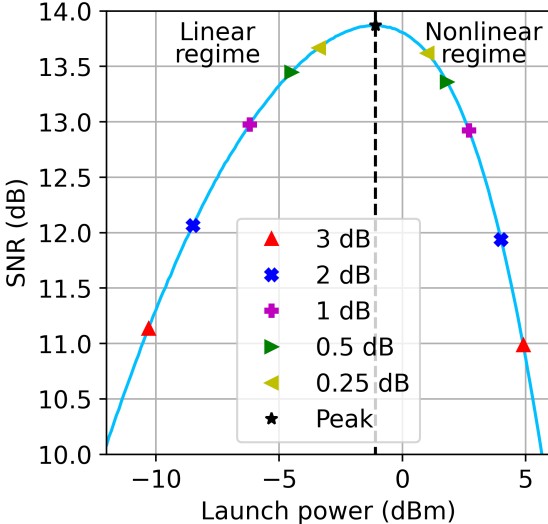

Figure 1: Simulated dataset of SNR vs launch power generated using the simulator, for SNR penalties of 0.25, 0.5, 1, 2 and 3 dB. Here the solid curve is included to show the behavior of the simulator at intermediate launch power values. The ground truth parameters used are $\alpha = 0.2$ dBkm$^{-1}$, $\gamma = 1.2$ W$^{-1}$km$^{-1}$, NF= 4.5 dB and $\text{SNR}_0 = 14.8$ dB. Also marked are the optimal operating point at -1.1 dBm, and the linear and nonlinear physical regimes.

the SNR that is obtained by connecting the transmitter directly to the receiver. $\text{SNR}_0$ describes the quantity of noise that is added to the signal by the transceiver. $b$ can be estimated using models of the nonlinear physics of transmission (Agrawal 2013). In Equation 3, as the launch power decreases $bP^3$ becomes small and $a$ dominates, meaning that SNR variation with launch power is linear, which we call the linear regime. At high power, $bP^3$ dominates and the SNR dependence on power becomes nonlinear, which we call the nonlinear regime. Thus, the launch power at which we measure changes the physical behavior of the system.

We utilize the expensive split-step Fourier method (SSFM) (Ip and Kahn 2008) in our simulator, as it is offers unparalleled accuracy. This allows us to estimate $b$ and thus to calculate SNR at a given launch power via Equation 3 using estimates for NF and $\text{SNR}_0$. Thus, the simulator takes as input a set of parameters pertaining to the characteristics of the system components, as well as the launch power.

## Simulated Dataset Generation

To demonstrate our method, we use the simulator with a set of ground-truth parameters, outlined in Table 1, to generate a dataset of SNR as a function of launch power, shown in Figure 1, and infer the set of ground truth parameters from this dataset. Specifically, we estimate the fiber attenuation coefficient $\alpha$, the fiber nonlinearity coefficient $\gamma$, the amplifier NF, and the transceiver back-to-back SNR $SNR_0$. The launch powers at which we simulate the SNR are chosen as those that correspond to an SNR penalty of 0.25, 0.5, 1, 2 and 3 dB, to a power precision of 0.1 dBm. Here, SNR penalty refers to the difference between a given SNR and the optimum SNR.

## Physical Layer Parameter Estimation Approach

---

Algorithm 1: Parameter estimation process

---

1) Let $X = \{X_i = \{x_1, x_2, \ldots, x_j, \ldots, x_m\} : j_L \leq x_j \leq j_U, 1 \leq i < \infty, 1 \leq j \leq m$ be the continuous sample space containing the samples $X_i$ consisting of a set of $m$ physical layer parameters $x_j$ with specified ranges bounded by upper and lower limits $j_U$ and $j_L$ respectively. Let $P_{\text{GPE}}$ be a set of launch powers, $X_{sol} \subseteq X$ be a solution set, $n_{sam}$ be the number of GPE training samples, $n_{HM}$ be the number of HM samples, and $L1, L2$ be the $L1, L2$ error norms with respect to the ground truth dataset respectively.
**for** power $p_j \in P_{\text{GPE}}$ **do**
  2) Train GPE$_j$ :
  **for** $k := [1, .., n_{sam}]$ **do**
    Draw sample $X_k := \text{LHD}(X)$.
    $\text{SNR}_{j,k} := \text{Simulator}(X_k, p_j)$.
  **end for**
  Optimize GPE$_j$ hyperparameters.
  Validate GPE$_j$.
  3) perform HM:
  Let $X_{sol_j} = \{\}$ be the set of plausible solutions for power $p_j$.
  **for** $i := [1, .., n_{HM}]$ **do**
    Draw sample $X_i := \text{LHD}(X)$.
    **if** $X_i$ is plausible based on Equation 2 **then**
      $X_{sol_j} := X_{sol_j} \cup X_i$.
    **end if**
  **end for**
  Round $X_{sol_j}$ to 3 significant figures
  4) $X_{sol} := X_{sol} \cap X_{sol_j}$.
**end for**
5) Generate GPE predictions for $X_{sol}$ at $P_{GPE}$:
**for** $r := [1, .., |X_{sol}|]$ **do**
  **for** $p_j \in P_{GPE}$ **do**
    $\text{SNR}_{j,r} := \text{GPE}_j(X_r, p_j)$.
  **end for**
**end for**
6) $X_{best} := \text{argmin}(L1, L2)$.

---

The proposed process for physical layer parameter estimation using GPE-driven HM is described in Algorithm 1. We draw 200 samples from the input parameter space of the simulation $X$ using a Latin hypercube design (LHD), for efficient coverage of the input space (Stein 1987). Table 1 shows the parameter ranges, chosen such that the ground truth parameters do not lie at the exact center of the ranges,

to ensure that the ground truth cannot be obtained via any averaging effects across the range. Then, we train a separate GPE for each launch power value, corresponding to SNR penalties of 0.25, 0.5, 1, 2 and 3 dB. The features of $X$ are the target physical layer parameters and a GP is trained on the simulator SNR predictions for $X$ to learn the variation of the SNR with the parameters. An additional 20 samples are drawn for validation of the trained GPE.

This process is then repeated for $n_p$ different launch power values, to learn the SNR variation with the parameters in the linear and nonlinear physical regimes. Following this, HM is performed and we generate SNR predictions from the trained GPE models for $n_{HM}$ LHD samples of the parameter space and compare them to the corresponding simulated SNR target using Equation 2. This process is repeated for $n_p$ separate launch power values, producing $n_p$ sets of candidate solutions $X_{sol_1}, X_{sol_2}, ..., X_{sol_{n_p}}$. The values of these parameters are then rounded to 3 significant figures. We then take the intersection $X_{sol1} \cap X_{sol_2} \cap ... \cap X_{sol_{n_p}}$ to produce a single set of candidate solutions $X_{sol}$. In doing this, we consider candidate solutions that are consistent with simulated data in the linear and nonlinear physical regimes, which allows us to narrow down the set of plausible parameters. To select the best set, we then input each set of candidate parameters into the trained GPE models to generate a set of SNR values at the target launch powers. These values are then compared to the corresponding data targets, and the optimal sets are selected as those for which the error vector minimizes the L1-norm and L2-norm. Here only $n_p$ launch power values have been used, and thus only $n_p$ measurements would be required to use this method for a deployed system. We consider two error metrics as each has a different qualities. The L1-norm is the simplest error measure to interpret, as it is simply the sum of the absolute value of the differences between the ground truth and the results being tested, and the L2-norm penalizes larger deviations more strongly than smaller ones. It should also be noted that practically, Algorithm 1 must be run link-by-link in a real network, as the physical layer parameters may vary spatially.

## Results

In order to validate the accuracy of the GPE models used, we draw an extra 20 samples from the parameter space using a LHD and evaluate the error of the GPE predictions with respect to the simulator. Figure 2 shows the mean of the L1 and L2 error norms across the 20 validation samples. Thus, 200 samples is sufficient for the GPE to learn the dependence of the simulator SNR output on the physical layer parameters to within a precision of at least 0.003 dB. This corresponds to a relative error of 0.03%, which provides empirical justification for the choices made in the design of the GPE approach.

In choosing the launch powers used for physical layer parameter estimation, there is a trade-off between minimizing the SNR penalty and probing further into the linear and nonlinear physical regimes, which will yield parameters that are consistent with all physical regimes and thus are more likely to be close to the ground truth. In optical networks, mea-

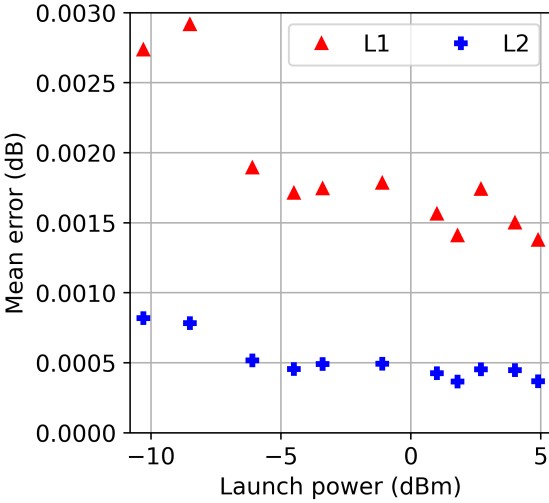

Figure 2: Mean of L1 and L2 norm errors with respect to simulator output for 20 GPE validation runs for each launch power used in the estimation.

Table 2: Physical Layer Parameter Estimates

| SNR penalty | $\alpha$ | $\gamma$ | NF | $\text{SNR}_0$ | $\overline{|X_{sol}|}$ |
|---|---|---|---|---|---|
| G. TRUTH | 0.200 | 1.20 | 4.50 | 14.8 | - |
| 3 dB | 0.200 | 1.20 | 4.50 | 14.8 | 271 |
| 2 dB | 0.200 | 1.20 | 4.50 | 14.8 | 551 |
| 1 dB | 0.200 | 1.20 | 4.49 | 14.8 | 1612 |
| 0.5 dB (L1) | 0.198 | 1.19 | 4.71 | 14.8 | 4426 |
| 0.5 dB (L2) | 0.198 | 1.19 | 4.70 | 14.8 | 4426 |
| 0.25 dB | 0.201 | 1.21 | 4.39 | 14.8 | 10642 |

parameters. However, the results with 1 dB are also highly accurate, with only a 0.2% error in NF. This interpretation is informed by the observation that the number of candidate solutions $\overline{|X_{sol}|}$, averaged over 5 HM runs, remaining after the intersection operation in step 4 of Algorithm 1 decreases as we increase the SNR penalty incurred. Therefore, as we move away from the optimum, we narrow the set of plausible parameters to those that are consistent with data from both the linear and nonlinear regimes, as well as the optimum, leading to a better estimation of the parameters.

## Conclusions and Future Work

In this work we have presented a novel algorithm for physical layer parameter estimation in optical fiber communication networks, based on GP-driven HM. As we wish to minimize the SNR penalty incurred by taking measurements, we investigated the trade-off between the SNR penalty and the quality of the estimation of physical layer parameters. Searching a broad parameter space, defined by a priori knowledge from typical network component specification sheets and the literature, we estimated a set of ground truth parameter values from simulated data. We found that as the SNR penalty increases, the quality of the parameter estimation increases. This is because at high SNR penalty, meaning launch powers far away from the optimum, we are using data from from far into the linear and nonlinear regimes. Thus, the parameters that are consistent with the data more accurately describe the linear and nonlinear regimes, leading to an improved parameter estimate. For a penalty of 2 dB or higher, the parameters were estimated precisely to 3 significant figures, while a 1 dB SNR penalty yielded an precise estimation of 3 of the 4 parameters, with only a 0.2% error in the NF. This method presents a way to improve the modeling of optical fiber networks, as it allows us to infer the parameters describing the behavior of the network components for any two connected nodes using measurement equipment that is installed as standard. In turn, this improves network design and facilitates virtual models such as digital twins. In future we aim to investigate the impact of system measurement noise and higher dimension parameter spaces on the efficacy of this method.

surements at non-optimal launch power values cause SNR penalties for services in the network, whereas taking measurements at the optimal launch power causes minimal disruption, assuming operation at the optimal launch power. We thus choose to use only $n_p = 3$ launch power values including at the optimal power, for SNR penalty thresholds of 0.25, 0.5, 1, 2 and 3 dB. A practical limit on $n_{HM}$ is enforced by the memory requirements of the arrays stored during HM. We used $n_{HM} = 1.9 \times 10^7$ for all results, which was the largest sample size we could use with the computing resources available. This was observed to be sufficiently large to ensure consistency across 5 HM runs for all launch powers considered.

Table 2 shows the results of the physical layer parameter estimation, where $X_{sol}$ is defined as in Algorithm 1. For an SNR penalty of 2 and 3 dB, the parameters are precisely estimated to the precision of 3 significant figures used. For 1 dB, all parameters except the NF are precisely estimated, for which the deviation from the ground truth is 0.2%. For a penalty of 0.5 dB, we see a different NF estimate depending on whether the L1 or L2 norm is used to select the optimal parameters, whereas for all other SNR penalties these norms yielded the same parameters. A parameter error of 1%, 0.8%, and 4.7% (L1) or 4.4% (L2) is observed for $\alpha$, $\gamma$, and NF respectively. $\text{SNR}_0$ is still precisely estimated. Finally, for 0.25 dB we see an error of 0.5%, 0.8%, and 2.4% for $\alpha$, $\gamma$, and NF respectively. The improved estimation for higher SNR penalty is caused by the fact that, as we move further from the optimal launch power, we are able to include information from further into the linear and nonlinear physical regimes, as described in Equation 3. Thus, the parameters that are compatible with the data as determined by HM are more likely to be close to the ground truth. For this specific simulator, we find that an SNR penalty of 2 dB is required to ensure precise estimation of the ground truth

## Acknowledgement
We thank the EPSRC for funding through TRANSNET (EP/R035342/1) and the IPES CDT (EP/L015455/1).

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

# Appendix: Glossary of Domain-Specific Terms

**Amplifier noise figure** (NF) A quantity that is directly proportional of the noise contribution of a given amplifier.

**Decibel-milliwatt** (dBm) A unit to express power level with reference to one milliwatt, commonly used to measure signal powers in optical networks.

**Fiber attenuation coefficient** ($\alpha$) A measure of how much a unit length of a given optical fiber attenuates an optical signal.

**Fiber nonlinearity coefficient** ($\gamma$) A measure of the strength of the nonlinear interactions between optical signals in a given optical fiber per unit length per unit optical power in the fiber.

**Launch power** The optical power with which modulated optical signals enter a span of fiber at the transmitter.

**Linear noise** Noise originating from the amplifiers that dominates when the launch power is small, parametrized by $a$ in Equation 3. For the EDFA amplifiers modeled, the dominant linear noise source is amplified spontaneous emission noise.

**Network monitors** Measurement equipment that is installed in a real-world optical network to monitor a range of metrics over time during the operation of the network, such as the SNR.

**Nonlinear noise** The contribution to the total noise caused by nonlinear interactions between laser signals in the optical fiber, which stems from the optical Kerr effect. This effect is parametrized by $b$ in Equation 3.

**Optical network** A network in which the vertices are comprised of optical transceivers and switches, and the edges are made up of spans of optical fiber, connected via in line optical amplifiers. Information is carried between nodes in the network using modulated laser signals.

**Optical Network Link** A connection between two nodes in an optical network, spanning a physical path through the network, over which data is transferred.

**Optical network physical layer** The first layer defined in the Open Systems Interconnection model (Zimmerman 1980), which concerns how raw bits are transmitted through an optical network, via the medium of a modulated laser. Parameters pertaining to this layer describe the physical behavior of network components.

**Quality of transmission** (QoT) A metric that quantifies the quality of a modulated laser signal, such as the signal to noise ratio.

**SNR penalty** The difference between the optimal SNR and the current SNR, which can be caused by using a non-optimal launch power.

**Split-step Fourier method** (SSFM) A method for estimation of the nonlinear effects in an optical fiber. This method works by splitting up the fiber into steps and solving the nonlinear Schrödinger equation iteratively, in order to model the propagation of the laser signal through the fiber (Agrawal 2013).

**Transceiver back-to-back SNR** ($SNR_0$) The SNR that is achieved by connecting the transmitter to the receiver, which is a measure of the contribution of the transceiver to the total noise.

## Appendix: Description of Optical Network Link Simulator

Here we present a more detailed description of the optical network link simulator used in this work. The simulator is designed to model a link consisting of a single channel transmitted using the quadrature phase-shift keying (QPSK) modulation format (Agrawal 2021) over 10 spans of length 100km. In this simulation, launch power is uniform across the spans and the signal is amplified by a 25 dB fixed-gain EDFA, with a variable optical attenuator (VOA) to compensate for the extra gain.

## Appendix: Details of Implementation and Simulation Set-up

The details of the implementation and simulation set up are described here. The simulator is implemented in MATLAB 2020 and with parallelisation enabled by MATLAB's GPU functionality. We use the MOGP emulator library (Daub 2021) implementation of the GPE model and HM routine, written in Python 3. As only uninformative priors have been provided, the GP kernel hyperparameters are selected by maximum likelihood estimation (Millar 2011), a special case of maximum a posteriori estimation with uniform prior distributions for the hyperparameters (Myung 2003; Daub 2021). This is performed by minimizing the negative likelihood using the SciPy implementation of the L-BFGS-B algorithm (Zhu et al. 1997). The simulations are run on using a single Nvidia P100 GPU with Intel Xeon E5-2650 v4 2.2GHz 12-core processors and 16GB memory. 200 training samples and 20 validation samples are drawn from the simulator for training of each GPE. HM is run on a CPU cluster with Intel Xeon Skylake 2.6GHz 16-core processors with 6840MiB memory per CPU, using 50 nodes.