# OpenReview forum: "Gaussian Process-Driven History Matching for Physical Layer Parameter Estimation in Optical Fiber Communication Networks"
_AAAI.org/2022/Workshop/ADAM — AAAI 2022 Workshop ADAM_

### Official Review · Reviewer_9hAQ · 2021-11-30
**Review of Gaussian Process-Driven History Matching for Physical Layer Parameter Estimation in Optical Fiber Communication Networks**

**Rating:** 7
**Confidence:** 5

**Review:**

This paper has demonstrated a GP-driven history matching (HM) approach for parameter estimation/calibration at the physical layer of a optical fiber comm network. The authors have laid out the motivation of this problem well and they have shown that this approach can yield a high accuracy in estimation with a low data requirement.
Pros: a suitable application of GP-based HM in parameter estimation; algorithms are clearly described in detail for reproduction by a practitioner.
Cons: Generally GP-based approach fails to scale with dimension. It is not clear from the paper how this approach would perform regarding accuracy and data requirement when the parameter space is high dimensional.

---

### Official Review · Reviewer_xWV2 · 2021-12-02
**Well written paper for GP based system ID for optical fiber communication networks**

**Rating:** 7
**Confidence:** 4

**Review:**

This paper presents a GP approach to create a data-driven emulatorof a detailed physics optical fiber communication network simulator. The GP approach naturally accounts for parameter uncertainty and serves as a robust surrogate model that can be used for parameter estimation based on observations. This is well-motivated problem.
Some suggestions/clarifications:-
- Clarification on how expensive the SSFM approach is to create the dataset
- Impact of synthetic noise on parameter estimation, specifically comment on the trade-off between dataset size and the maximum underlying noise that the estimator can handle.